# Real-space imaging with pattern recognition of a ligand-protected Ag$_{374}$ nanocluster at sub-molecular resolution

Qin Zhou[1,2], Sami Kaappa [3], Sami Malola[3], Hui Lu[2], Dawei Guan[2], Yajuan Li[2], Haochen Wang[2], Zhaoxiong Xie[1], Zhibo Ma[2], Hannu Häkkinen [3], Nanfeng Zheng [1], Xueming Yang[2,4] & Lansun Zheng[1]

High-resolution real-space imaging of nanoparticle surfaces is desirable for better understanding of surface composition and morphology, molecular interactions at the surface, and nanoparticle chemical functionality in its environment. However, achieving molecular or sub-molecular resolution has proven to be very challenging, due to highly curved nanoparticle surfaces and often insufficient knowledge of the monolayer composition. Here, we demonstrate sub-molecular resolution in scanning tunneling microscopy imaging of thiol monolayer of a 5 nm nanoparticle Ag$_{374}$ protected by *tert*-butyl benzene thiol. The experimental data is confirmed by comparisons through a pattern recognition algorithm to simulated topography images from density functional theory using the known total structure of the Ag$_{374}$ nanocluster. Our work demonstrates a working methodology for investigations of structure and composition of organic monolayers on curved nanoparticle surfaces, which helps designing functionalities for nanoparticle-based applications.

[1] State Key Laboratory of Physical Chemistry of Solid Surfaces, Collaborative Innovation Center of Chemistry for Energy Materials,and National & Local Joint Engineering Research Center for Preparation Technology of Nanomaterials, College of Chemistry and Chemical Engineering, Xiamen University, Xiamen 361005, China. [2] State Key Laboratory of Molecular Reaction Dynamics, Dalian Institute of Chemical Physics, Chinese Academy of Science, Dalian 116023, China. [3] Departments of Physics and Chemistry, Nanoscience Center (NSC), University of Jyväskylä, FI-40014 Jyväskylä, Finland. [4] Department of Chemistry, Southern University of Science and Technology, 1088 Xueyuan Road, Guangdong, Shenzhen 518055, China. Correspondence and requests for materials should be addressed to Z.M. (email: zhbma@dicp.ac.cn) or to H.Häk. (email: hannu.j.hakkinen@jyu.fi) or to N.Z. (email: nfzheng@xmu.edu.cn)

Real-space imaging of nanoparticles is instrumental for understanding the correlations between size, shape, composition, structure, and stability. These fundamental properties define functionalities for applications in areas, such as catalysis, drug delivery, bio-sensing, and theranostics[1,2]. Most metal and semiconductor nanoparticles are made by wet chemistry using surface-stabilizing molecules that build an organic overlayer on the nanoparticle surface; consequently, knowing the detailed properties of this overlayer is crucial for understanding how nanoparticles interact with their environment[3]. While great progress has been made in imaging organic molecules on flat surfaces via atomic force (AFM) and scanning tunneling (STM) microscopies, with resolution extending down to a single intramolecular covalent bond[4–9], achieving even a molecular resolution at nanoparticle organic surface has proven to be extremely challenging[10–17]. These challenges arise from the highly curved nanoparticle surface leading to spatially dependent tip-convolution, inhomogeneity of nanoparticle size and shape, insufficient knowledge of the composition of the nanoparticle organic surface, and from the difficulty to produce accurate theoretical modeling for AFM or STM imaging due to lack of reliable atomic-scale nanoparticle models.

Atomically precise metal nanoclusters, composed of 1–3 nm noble metal cores and an organic monolayer of thiols, phosphines, or alkynyls, have emerged in recent years as an interesting sub-class of extremely well-defined nanoparticles with total structures available from single crystal X-ray crystallography[18–20]. Recently, synthesis and the total structure of a large silver nanocluster of composition $Ag_{374}(SR)_{113}Br_2Cl_2$ (SR = SPhC$(CH_3)_3$ = tert-butyl benzenethiol, TBBT) was reported[21]. The $Ag_{374}$ nanocluster has a decahedral silver core of about 3 nm in diameter, and the total diameter of the particle is about 5 nm (Fig. 1a, b). Density functional theory (DFT) calculations on this cluster indicated a metallic behavior with a continuous density of electron states at the Fermi level, and experimental UV–Vis spectroscopy revealed a strong plasmonic absorption at 465 nm (2.67 eV)[21]. The $Ag_{374}$ nanocluster serves thus as a model of a large metallic nanoparticle, yet structurally known to atomic precision.

Here, we show that it is possible to achieve a sub-molecular spatial resolution in STM imaging of topography of the TBBT surface of $Ag_{374}$ nanoclusters in ultra-high vacuum (UHV) conditions at both liquid Helium (LHe) and liquid Nitrogen (LN$_2$) temperatures. Spatial recognition of single $CH_3$ units of SPhC$(CH_3)_3$ at the nanocluster surface is corroborated by comparisons to a set of STM images of individual TBBT molecules at various orientations on Au(111) surface, as well as to DFT simulations of topography images based on the known atomic structure of $Ag_{374}(SPhC(CH_3)_3)_{113}Br_2Cl_2$. We demonstrate, provide, and discuss an algorithm to automatically suggest the closest matches between simulated STM images and the experimental data, based on principles in "facial recognition".

## Results

**Nanocluster synthesis and characterization.** The $Ag_{374}$ nanoclusters were synthesized and characterized (Supplementary Fig. 1) as described previously[21], deposited on dithiol-modified Au(111) surface at ambient conditions and quickly transformed to a UHV-STM chamber (see Methods for details). The clusters formed a disordered monolayer as shown in Fig. 1c. The particles show almost the same apparent height (about 3 nm) but present rather different widths and shapes induced by tip convolution effect depending on the surrounding as well as their orientations. We applied a simple method to extract the real nanocluster diameter from STM images. First, we located an area containing

an ordered compact nanocluster array with at least three particles at the same altitude, as shown in Fig. 1c. Second, each $Ag_{374}$ was identified (indicated by blue dashed circles in Fig. 1c). Excluding the last particle of the array, the height profile along the row of clusters exhibits regular oscillations at about 5 nm intervals which is taken as the true size, free from convolution effects (red line, profile (1), in Fig. 1d). This size corresponds well to the cluster diameter determined from the crystal structure. On the other hand, height profile taken across the cluster line indicates cluster diameter of about 10 nm; this value includes the tip convolution effect (black line, profile (2), in Fig. 1d).

We found that continuous scanning of the same sample area improved both the sample stability and the spatial resolution (Fig. 2). We started with higher setpoint currents of 40–30 pA for a typical bias voltage of −1.2 V (meaning that the STM tip scans closer to clusters) and eventually reduced the setpoint current to 10 pA. During the scans at higher currents, we noticed movement of less stable clusters away from the sample area (removed by the tip), as well as gradual increase of the spatial resolution of topography features at the cluster surfaces. With the final setpoint values of −1.2 V and 10 pA, the images were stable at 20 min time scale over a few scans. This pertained to the individual topography features at the same cluster surface to a very high degree independent of the scanning directions (Supplementary Fig. 2).

**DFT calculations and STM topography simulations.** Large-scale DFT calculations by using the GPAW software[22] were made to solve the ground-state electron density of $Ag_{374}(SPhC(CH_3)_3)_{113}Br_2Cl_2$ in its determined X-ray structure[21], and the Tersoff–Haman method[23] was used to simulate STM topography of $Ag_{374}$'s organic surface (see Methods for details). Representative DFT-topography image and height profile across the cluster surface, calculated using typical bias voltage and current values of the experiment, are shown in Fig. 3a, b, respectively. DFT gave an apparent size of one TBBT ligand as 0.6 nm and indicated that each methyl group in the ligand is visible, with peak-to-peak distance of about 0.3 nm. Imaging of single TBBT molecules adsorbed on Au(111) surface provided valuable additional information of various modes of appearance (Supplementary Fig. 3). As shown in Fig. 3c, in most cases only one or two of the methyl groups were seen due to the tilt angle between TBBT and the surface, but the peak-to-peak distance was in the range of 0.3–0.4 nm (Fig. 3d), consistent with the DFT results in Fig. 3a, b.

**Confirmation of sub-molecular resolution.** Encouraged by these observations, we turned to systematic analysis of several high-resolution STM images of $Ag_{374}$ taken at LHe temperature; a typical example is shown in Fig. 3e–h. As shown in Fig. 3e, the topography at the center area of the cluster, where tip convolution effects are minimal, featured rather regular maxima and minima, with peak-to-peak distances ranging from 0.3 to 0.6 nm (Fig. 3f). Concentrating on 11 locations at the cluster surface, we were additionally able to identify local configurations of maxima/minima that closely resemble individual TBBT molecules on the Au(111) surface (Fig. 3g, h). All this evidence points unambiguously that single methyl groups produced the topography variations in our image data, i.e., we achieved the sub-molecular resolution.

We did a similar analysis of $Ag_{374}$ topography data taken at LN$_2$ temperature (Supplementary Figs. 4–6). Comparison of images taken from the same sample area during continuous scanning showed fewer sharply identifiable ligands and somewhat less sharp peaks in the height profile compared to data measured at LHe temperature, as expected. However, the main features of

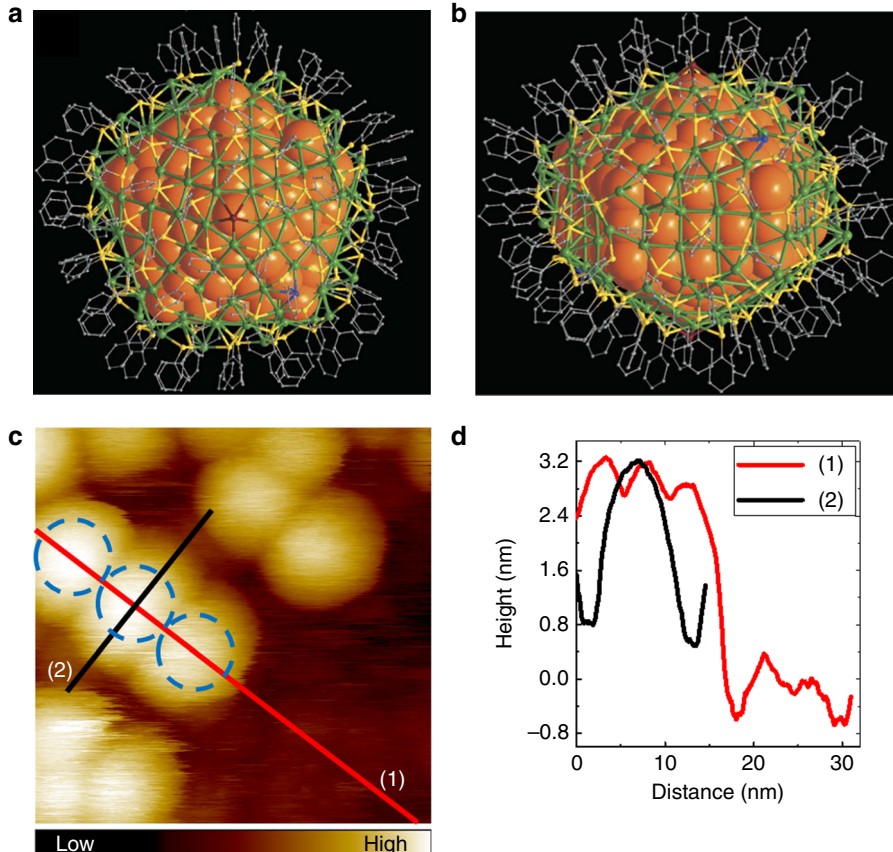

**Fig. 1** Total structure of Ag$_{374}$ nanoparticle and its appearance in STM imaging. **a**, **b** Complete cluster structure shown in a plane perpendicular to a five-fold rotation axis and a plane parallel to the five-fold axis, respectively (from ref.[21], copyright 2016 SpringerNature). **c** STM topography image of Ag$_{374}$ at LHe temperature. Scan size: 25.1 × 25.1 nm$^2$, bias and setpoint current: 1.2 V and 30 pA. Blue dashed circles indicate nanoclusters with a defined size (diameter 5 nm) and their locations. **d** Height profiles taken along the red and black lines in **c**

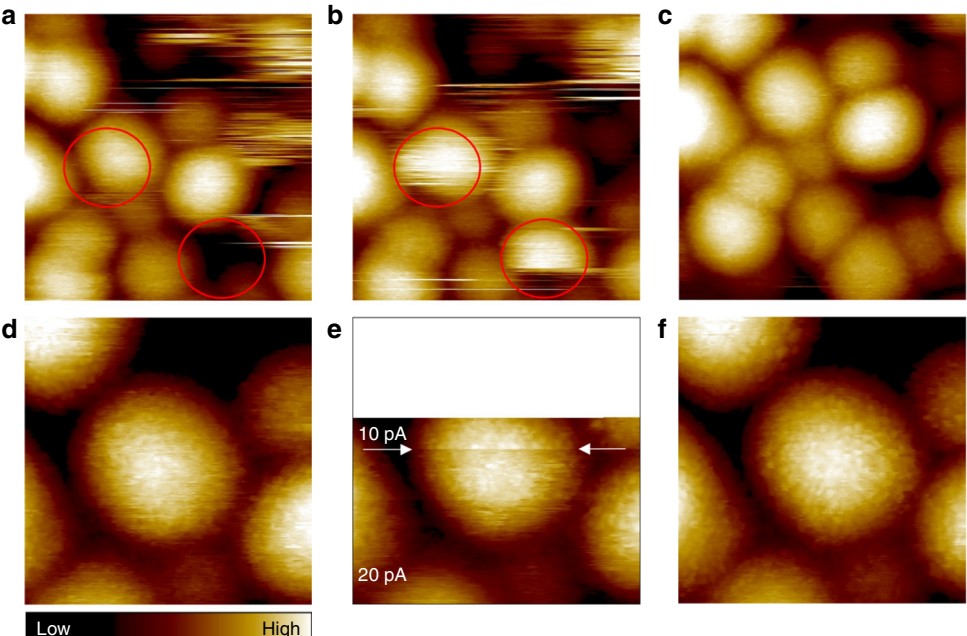

**Fig. 2** Repeated imaging of the same sample area increases the resolution. Bias, setpoint current, scan size: **a**, **b** −1.2 V, 30 pA, 23.7 × 23.7 nm$^2$ **c** −1.2 V, 20 pA, 23.7 × 23.7 nm$^2$ **d** −1.2 V, 20 pA, 12.0 × 12.0 nm$^2$ **e** −1.2 V, setpoint current change from 20 to 10 pA at the marked scan line, 12.0 × 7.8 nm$^2$; and **f** −1.2 V, 10 pA, 12.0 × 12.0 nm$^2$. Red circles indicate unstable clusters and some were removed by tip in in **a**, **b** . Repeated imaging of the same area with decreasing setpoint current improved the spatial resolution markedly. Images taken at the LHe temperature over the course of 7 min/image

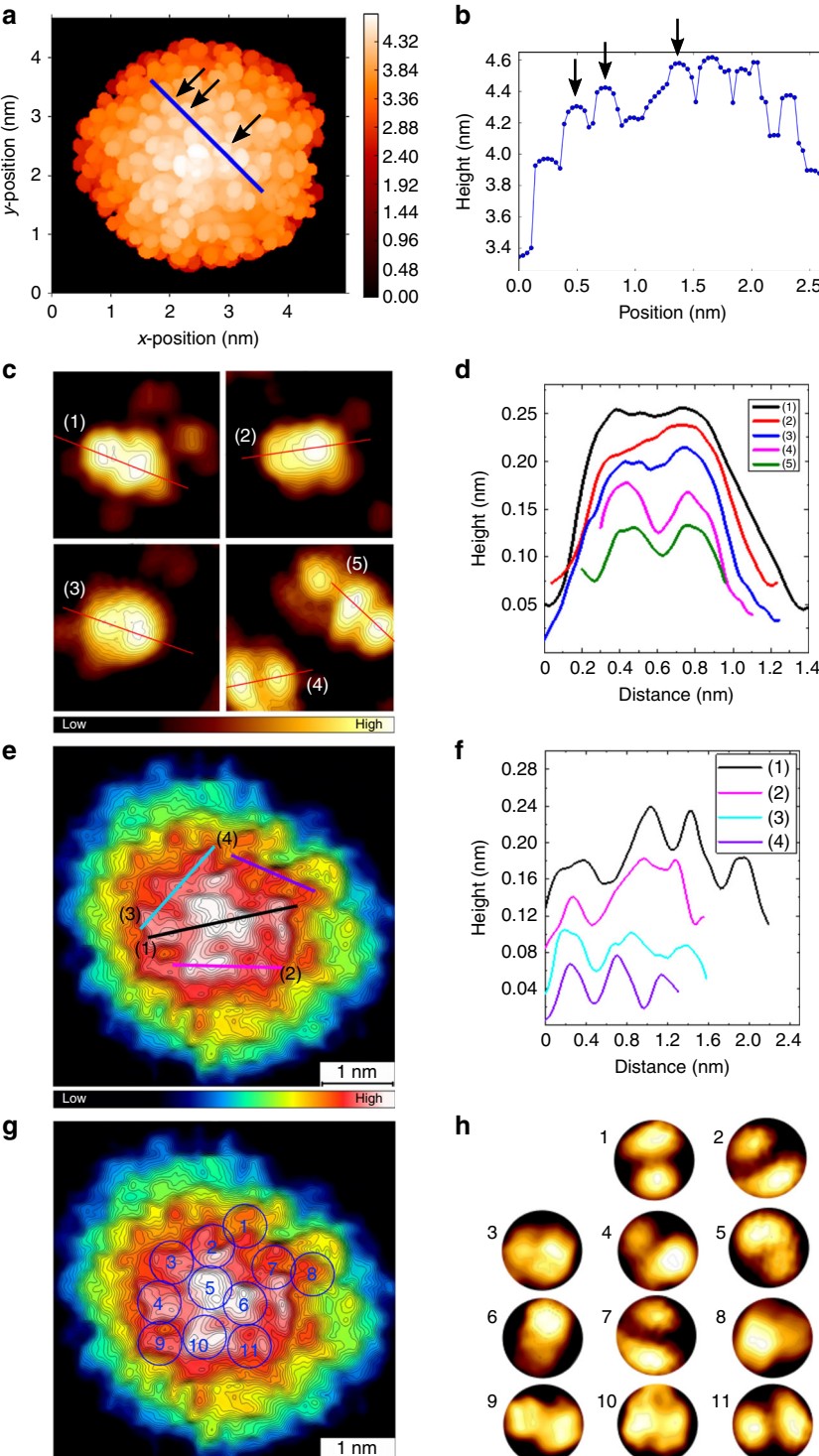

**Fig. 3** DFT simulations and high-resolution imaging. **a** DFT calculated STM image for $Ag_{374}$ by using the known atomistic structure of the nanocluster[17]. Image area: $5 \times 5 \ nm^2$. The image has been simulated by using voltage and current settings of $-1.4$ V and 20 pA, respectively. Black arrows indicate individual methyl groups. **b** Height profile taken along the blue line in **a**. The three arrows show positions of the same methyl groups as shown in **a**. **c** STM topography images of single TBBT molecules in various orientations bound to an Au(111) surface. Image settings: (**1**,**2**) 1.0 V, 30 pA, $1.51 \times 1.51 \ nm^2$; (**3**–**5**) 1.2 V, 60 pA, $1.50 \times 1.50 \ nm^2$, all at $LN_2$ temperature. (Zoomed in from different large images, the resolution had small deviation.) **d** shows height scans of the molecules as marked in **c** by numbers 1–5. **e** A topography image of $Ag_{374}$ at submolecular resolution (bias: $-1.2$ V, current: 10 pA, scan size: $4.81 \times 4.81 \ nm^2$). **f** Height profiles of the cluster along the lines marked in **e**. **g** The same cluster as in **e**, with 11 detailed features showing similarity to the ligand configurations at Au(111) shown in **h**. STM imaging at LHe temperature

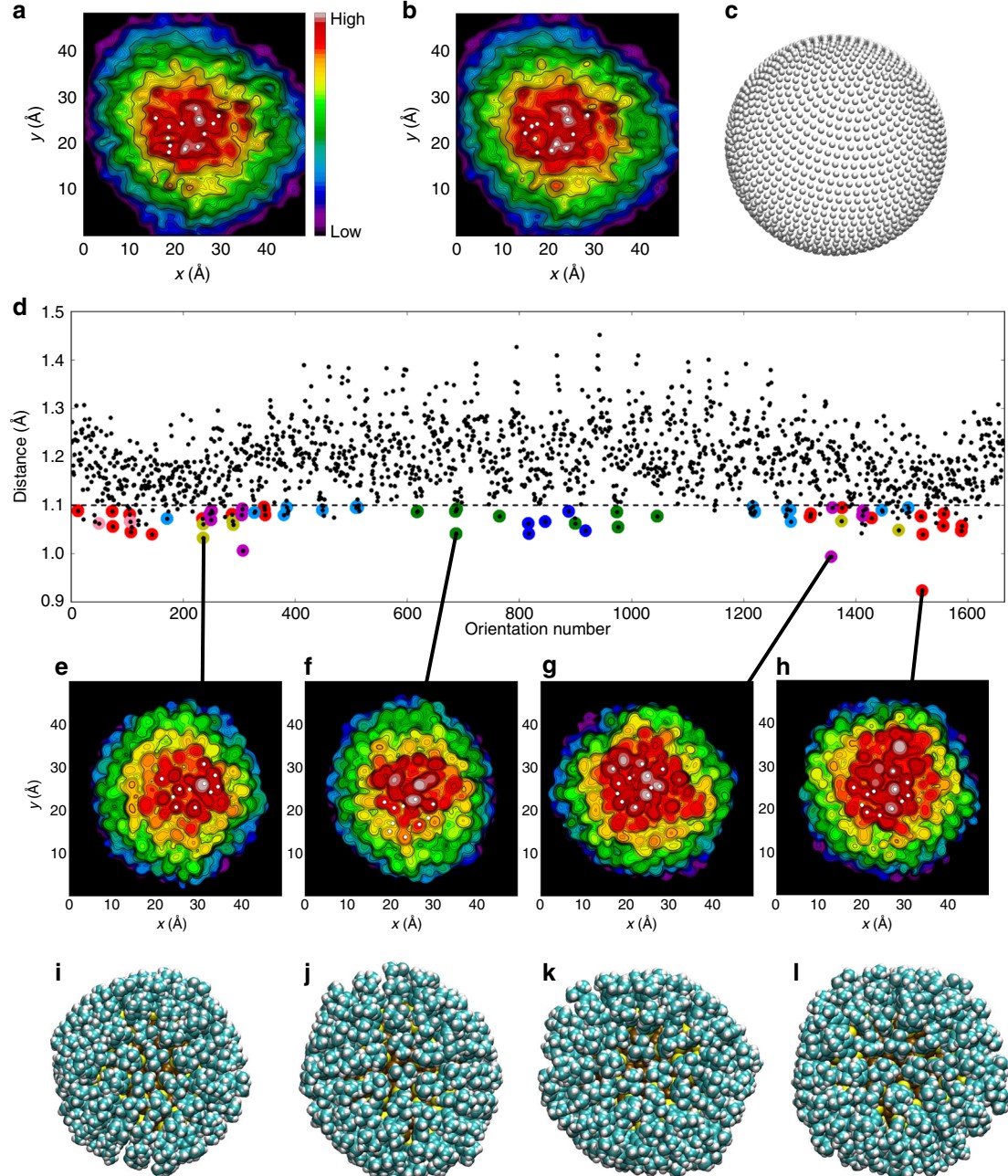

**Fig. 4** "Facial recognition" of experimental STM topography by an automated matching of simulated images. **a**, **b** Two consecutive scans (trace and retrace) of the same Ag$_{374}$ cluster at LHe temperature (bias voltage: −1.2 V, current: 10 pA). The white dots represent the extrema of the data. **c** Visualization of the set of perspectives used to prepare the simulated STM image from the computational data. Each dot corresponds to a perspective when the cluster lies at the center of the sphere. **d** Minimum correlation distances for each perspective, resulting from comparison to the extremum points of **a**. The dashed line is drawn below the major feature formed by the data points. Below this threshold, the data points with the same color correspond to a perspective very close to each other that consequently give relatively small correlation distances. Selected small-distance perspectives are connected with lines to the respective calculated STM images (**e–h**) with white dots showing the extrema. **i–l** The atomistic models of the cluster in similar orientations to **e–h**, respectively. X–Y scales are the same in experimental and simulated images, and are shown in Å. The STM simulations were done for the voltage and current values used in the experiment

the topography remained stable showing that sub-molecular resolution could be achieved.

**Pattern recognition between experimental and simulated topographies.** The sub-molecular resolution in the experimental topography data encouraged us to go one step further in comparison between experiment and theory, by applying an idea of

"facial recognition". By projecting the DFT-calculated local density of electronic states of Ag$_{374}$ to 1665 different space orientations (Fig. 4c), we calculated 1665 simulated topography images and compared them to experimental data (Fig. 4a), looking for the best match with help of an automated algorithm that used ideas for pattern recognition in machine learning. We determined a generalized rms-distance between each simulated topography image and experimental data based on locations of extrema

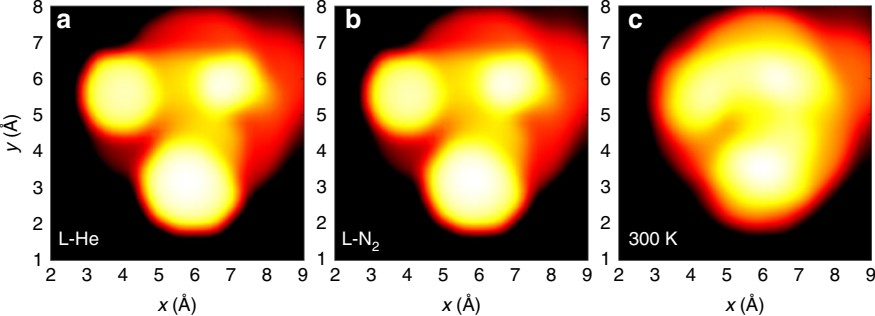

**Fig. 5** Temperature effect on simulated STM topography images of a single TBBT molecule on Au(111) surface. The images show the simulated STM topography images by taking into account the ensemble effect of various positions of the t-butyl group at LHe (**a**) and LN$_2$ temperatures (**b**), and at 300 K (**c**). The ensemble averages are calculated in each case as a weight-averaged of 20 images corresponding one 120° rotation from one potential minimum to the next one, separated by the energy barrier of 80 meV, the weight factors coming from Boltzmann term exp($-E/kT$). The averaged positions of single methyl groups are clearly resolved at LHe and LN$_2$ temperatures but are blurred at 300 K

points in the central region of the image (where the experimental tip convolution effects are minimal). Minimal values of the generalized distance were used as a criterion to find the best candidates in the simulated images. As Fig. 4d shows, only a handful of good matches out of 1665 candidates were identified by this algorithm, shown in Fig. 4e–h with the corresponding atomic visualizations in Fig. 4i–l. A similar result was achieved when we analyzed the STM topography data achieved by re-tracing (Fig. 4b) the same Ag$_{374}$ cluster as shown in Fig. 4a: a small sub-set from the possible 1665 orientations stands out as giving a close correlation to the experimental data (Supplementary Fig. 7). Self-consistent tests of the reliability of the matching algorithm, performed by taking one simulated image (out of 1665) as a reference ("experimental") data, adding random noise to the data and comparing that data to all other 1664 possible simulated images, showed that the true image is found while the random noise is kept within reasonable bounds (Supplementary Fig. 8).

As Fig. 4 shows, the closest matches between the "best" simulated and experimental topography images yield a typical rms-value of 1 Å for the distance (that is, an extremal point in the best simulated image is on average within 1 Å from a corresponding point in the experimental data). As the simulations pertain to strictly one ($T = 0$) configuration of the Ag$_{374}$ cluster, it is natural to question the role of thermal dynamics of the ligand layer in producing noise to the experimental data in the time scale of imaging. To address this question, we calculated by DFT energy barriers for three different modes of motion of a single TBBT ligand on a flat Au(111) surface: (i) rotation of the tert-butyl group around the C–C bond, (ii) rotation of the phenyl ring around the S–C bond, and (iii) flipping motion of TBBT with respect to an underlying Au–Au bridge of the surface. The results are summarized in Supplementary Fig. 9. We found that the rotational barriers are very low for mechnisms (i), (ii) described above (80 and 40 meV, respectively), while the barrier for the flipping motion is large, 0.27 eV. In order to apply an Arrhenius-type estimate for the rate of rotational motion at various temperatures, we took a typical rotational frequency of 10$^9$ 1/s considering the tert-butyl group/phenyl group as a rigid rotor. This implies that at the LHe temperature, even the rotational motion with the lowest barrier (rotation (ii), 40 meV) is effectively hindered (rate of the order of 10$^{-42}$ 1/s). Rotations can easily take place at the LN2 temperature during the imaging timescale (rates of 10$^3$–10$^6$ 1/s) and certainly at the room temperature, where the initial handling of the nanoparticle sample takes place. At room temperature, a flipping motion can in principle contribute as well, although in the tightly packed

TBBT monolayer of Ag$_{374}$ its energy barrier is probably significantly larger than shown here for an isolated TBBT/Au (111). Rotations give an average uncertainty of 0.8 Å in the rms-distance over one 120° rotation from one potential energy minimum to the next one (Supplementary Fig. 9e, f). We also studied the ensemble effect of rotations on the simulated STM images by weight-averaging 20 STM images of evenly spaced rotational configurations around the 120° rotation, by using the Boltzmann weight factors exp($-E/kT$). The results in Fig. 5 show interestingly, that single CH$_3$ groups still produce distinct intensity maxima also at LN$_2$ temperature, but at 300 K the sub-molecular resolution is blurred since the high-energy configurations are thermally populated by significant weights.

## Discussion

Previously, successful STM imaging of individual metal atoms[24–26], small metal and metal-oxide clusters[27,28], and metal nanoparticles at near-atomic resolution[29] on well-prepared flat supports have been reported. Hybrid nanoparticles, consisting of metal core and organic ligand layer, have been notoriously challenging to image to high spatial resolution due to the highly curved surface and uncertainties in the ligand monolayer composition. Our work demonstrates a successful approach for investigations of the structure of organic monolayers on curved nanoparticle surfaces at sub-molecular spatial resolution, by combining low-temperature STM imaging with high-level DFT simulations of STM topography data and automated comparison of large simulated data sets to experimental data via an algorithm based on pattern recognition. We believe that this approach will facilitate various studies of physical and chemical properties of ligand monolayers, such as conductance, ligand–ligand interactions, and chemical reactivity with the environment, at an unprecedented level of spatial resolution. The improved understanding of composition, morphology, and functionalities of ligand layers of hybrid nanoparticles will help designing nanoparticles for applications.

## Methods

**Nanocluster synthesis**. Ag$_{374}$ nanoclusters were prepared and crystallized as reported in ref.[17]. The crystals were stored at 4 °C in fridge to keep stable. 4,4′-Biphenyldithiol (Alfa, 97%) and 4-(t-butyl)phenylthiol (Alfa, 97%) were used without further purification. Ethanol (HPLC) and dichloromethane (HPLC) were used as solvent. 1 M HCl and 0.5 M H$_2$SO$_4$ were prepared by directly diluting concentrated HCl (37%) and H$_2$SO$_4$ (98%), respectively, to electrochemically polish Au(111).

**Au(111)/dithiol/Ag$_{374}$ sample preparation**. Au(111) substrate (MaTeck, 5 × 5 cm$^2$) was cleaned by Ar$^+$ sputtering (at 1.5 keV and 10$^{-6}$ mbar) and

annealing (at about 700 K) in UHV chamber after electrochemical polish in the air. Dithiol modified Au(111) was achieved by immersing the substrate to 1 mM 4,4′-Biphenyldithiol/$CH_3CH_2OH$ solution at 60 °C for 4 h in a sealed vessel isolated from light and cleaned with pure ethanol for several times. Then 5 μL $10^{-7}$ M $Ag_{374}/CH_2Cl_2$ solution was dropped to the dithiol-modified Au(111) substrate. After the solvent automatically and slowly evaporated from the surface, the sample was transferred to UHV-STM chamber.

**Au(111)/4-(t-butyl)phenylthiol sample preparation**. Clean Au(111) substrate was immersed in the atmosphere of 4-(t-butyl)phenylthiol in a sealed vessel at 60 °C for 1 h. Then it was transferred to UHV-STM chamber.

**STM measurements**. STM measurements were performed with a low-temperature STM system (Scientaomicron), operating at a base pressure of $10^{-11}$ mbar. The sample temperatures were either 7 K (LHe) or 79 K (LN₂). Tungsten tip was electrochemically etched in 5 M NaOH solution. Image processing was performed by SPIP 6.6.0 software.

**DFT calculations and STM topography simulations**. The electronic structure and STM topography images were calculated using the DFT as implemented in the real-space code-package grid-based projector augmented-wave method (GPAW)[22]. The experimentally determined[21] crystal structure of $Ag_{374}[SPhC(CH_3)_3]_{113}Br_2Cl_2$ was used directly without atomic relaxation. The ground state electron density was solved by using 0.25 Å grid spacing and the Perdew–Burke–Ernzerhof (PBE) xc-functional[30]. STM topographs were calculated from the partial local density of states by using the Tersoff–Hamann method[23]. 1665 orientations were used, distributed evenly with 5° separation either in azimuthal or polar direction. Bias voltage and tunneling current were selected according to the experimental setups.

DFT calculations on TBBT molecules on Au(111) surface were done in a x–y periodic supercell of four Au layers in the z-direction and 64 atoms in the cell, with one TBBT molecule relaxed at an initial Au–Au bridge position when the bottom Au(111) was fixed. The energy calculations were done with the gamma-point approximation. PBE functional and 0.20 Å grid spacing were used. Rotations of either the tert-butyl headgroup around the C–C bond or the phenyl group around the S–C bond were considered (Supplementary Fig. 9a, b). In addition, an energy barrier for a motion that flips the molecular axis to a symmetrically identical position around the surface Au–Au bridge (Supplementary Fig. 9c) was studied using the nudged elastic band method[31] with nine intermediate images and the spring constant of 0.1 eV/Å, and four closest Au atoms to S being dynamic.

**Numerical comparison of STM images for pattern recognition**. For each calculated STM image, we determined the coordinates of the 20 largest local maxima and 20 largest local minima after Gaussian smoothing of the data with standard deviation of 0.23 Å. The smoothing was performed in order to obtain more well-defined gradients of the numerical data and thus to model the experimental data in a more sophisticated way considering the local extremum points. A point was accepted as a local maximum (minimum) if it had the maximum (minimum) value of the data within a radius of 0.7 Å. Similarly, we determined a few well-defined extrema from the experimental data (all of the extrema are shown in Fig. 4a, b in the main text). We concentrated the analysis to the central part of the cluster due to aberrations near the cluster edges caused by the tip-convolution effect.

We compared an experimental set of extremum coordinates to each computational set as follows. The extremum coordinates were first placed into the same coordinate system by setting the maximum height coordinates to origin. The nearest neighbor for each experimental coordinate was sought from the computational set, separately for local maxima and minima. Denoting the set of extremum coordinates from the experimental data as Q and the nearest neighbors from the calculated data as P (correspondingly ordered), the rotation and translation matrices R and T, respectively, were sought that minimize the root-mean-square distance between the sets Q and P′ = RP + T, using the algorithm introduced in ref.[32]. The fitted distance between the coordinate sets Q and P is thus read as

$$D = \left( \Sigma_i |P'_i - Q_i|^2 / N \right)^{1/2} \quad (1)$$

where the summation index i runs from 1 to N that is the number of coordinates in Q. The distance D was minimized with respect to rotations of the experimental set of points around full circle to consider all the reasonable nearest-neighbors between the sets of coordinates. These minima are documented in Fig. 4d for each calculated image, compared to one of the experimental images.

**Code availability**. For DFT calculations and system preparation we used open-source software available at https://wiki.fysik.dtu.dk/gpaw/. The custom-made computer code used in comparison of simulated and experimental STM images can be downloaded from the web link http://r.jyu.fi/uNF.

## Data availability

All the data of this work is available from the corresponding authors (for experimental data, by request to zhbma@dicp.ac.cn or nfzheng@xmu.edu.cn and for computational data, by request to hannu.j.hakkinen@jyu.fi).

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

## Acknowledgements

The experimental work done in Dalian Institute of Chemical Physics (DICP), Chinese Academy of Sciences, was supported both by Xiamen University (The National Key R&D Program of China grant 2017YFA0207302, National Natural Science Foundation of China, grant 21731005, 21420102001 and 21721001 the National Key R&D Program of China grant 2017YFA0207302) and DICP (National Natural Science Foundation of China grant 21688102, the Strategic Priority Research Program of Chinese Academy of Science, grant XDB17000000, the National Key Research and Development Program of the MOST of China, grant 2016YFA0200603 and the open fund of the state key laboratory of molecular reaction dynamics in DICP, CAS, grant SKLMRD-K201707). Q.Z. thanks Dr. Huayan Yang for providing the samples for STM imaging. S.M. and H.H. thank T. Kärkkäinen and P. Nieminen for discussions on pattern recognition algorithms. The theoretical and computational work in the University of Jyväskylä was funded by the Academy of Finland (grants 294217, 315549, AIPSE program, and H.H.'s Academy Professorship). H.H. acknowledges the support from China's National Innovation and Intelligence Introduction Base visitor program. S.K. thanks the Vilho, Yrjö, and Kalle Väisälä Foundation for the grant for doctoral studies. The DFT simulations were done at the Finnish national supercomputing center CSC and at the Barcelona Supercomputing Center (PRACE project "NANOMETALS").

## Author contributions

N.F.Z. and L.S.Z. designed the study and supervised the project. Q.Z. carried out the STM experiments and analyzed the STM data, assisted by H.L., D.W.G., Y.J.L. and H.C.W. Z.B.M. and X.M.Y. supported the experiment and Z.X.X. supported the STM data analysis. S.M. carried out the DFT calculations of the electronic structure of Ag$_{374}$. S.K. carried out the STM topography simulations and built the algorithm of pattern recognition for comparison to experimental topography data. H.H. supervised the theoretical and computational work and analyzed the computational results together with S.K. and S.M. The first manuscript draft was compiled by Q.Z., H.H. edited and finalized the manuscript based on comments from co-authors.

## Additional information

**Competing interests:** The authors declare no competing interests.

