## [Peer Review File · Nature Communications]

Reviewers' comments:

Reviewer #1 (Remarks to the Author):

This paper describes the scanning tunnelling microscopy imaging (STM) of a silver cluster (Ag₃₇₄). To the best of my knowledge there is no report of such things to date. As such I think this paper should be published but in a specialised journal or a broad journal with reporting functions as Scientific Reports.

The significance of this paper as such is minimal. The STM does not add anything to what X-ray crystallography has shown so far also on the same cluster. In fact it requires strong support from calculation.

The STM itself has been widely published on nanoparticles by the group of Stellacci. The authors do reference a controversy there but do not comment on it and neither add to it. The approach used is almost identical to the one described by such group in Ong et al ACS Nano (non cited), and the results achieved are similar to the ones described in the same paper and in Moglianetti et al. (not cited). Their minimal difference is that they achieved these results in liquid nitrogen and helium temperature, but low temperature results were described in Biscarini et al. (not cited).

Given the scant discussion in the paper (lacks any point) and the two major objections report, I suggest rejection.

Reviewer #2 (Remarks to the Author):

The present manuscript presents a study that combines experiment and atomic scale modeling to resolve the origin of features on a submolecular length scale in the experimental topography maps of Ag nanoparticles. The study is well motivated and addresses a topic that is of interest for a wide research community. The approach itself combines state-of-the-art experimental and computational methods in a novel fashion.

I think this work could be suitable for Nature Communications if the authors can comprehensively address the following issues:

* I read the manuscript three times. Only upon the second reading did I see the main point. Hence I think the message should be sharpened still. For example it would help to avoid vague terms such as "facial recognition", which do not carry a particular meaning (at least not on first reading).

* The term "database" is used in multiple places. I assume this refers to the 1665 different orientations for which STM images for calculated? The authors also allude to a set of data for "flat" surfaces. It is unclear to me how these hang together.

* "Pattern recognition" is a wide field in machine learning that has produced a wealth of sophisticated algorithms, many of which are readily available via well maintained libraries. By comparison the algorithm employed here is quite simplistic. There is for example no statistical analysis of the reliability of the predictions. Moreover the minima in the generalized distance plotted in Figure 4D are not very pronounced. Hence, I am concerned how transferable this analysis is and how reliable the thus identified structures ought to be. What do I actually learn when looking at Figures 4I-L?

* Finally, while the measurements are carried out at very low temperatures, the molecules investigated here still feature rather soft rotational modes, which could in principle be populated at these temperatures. The "final" STM images are obtained by compounding multiple measurements over a longer period of time. How much molecular motion/vibration/rotation can I expect at these low temperatures over such long time scales and how would that affect the results? I would feel more confident if the authors could include a discussion of thermal effects and some estimates along these lines.

Reviewer #3 (Remarks to the Author):

The manuscript entitled "Real-Space Imaging and "Facial Recognition" of a Ligand-Protected Ag₃₇₄ Nanocluster at Sub-Molecular Resolution" by Häkkinen et al. provides an interesting tool to characterise metal nanostructures. They combined STM and computer simulation to characterise the Ag₃₇₄ particle. Both results, experimental and computational, were implemented in a mathematical algorithm to identify the facets even though ligands surrounded the particle. Although the results for facial recognition are not fully conclusive, the idea is original, and I believe it can lead to fast advances in the field of nano-science. The information provided in the manuscript and supplementary material is not enough to support the claims of "facial recognition". It can be, perhaps, that the simulation model is not accurate and therefore the points founded in the facial recognition step do not entirely agree with the experiment. The tip of the STM may also influence the arrangement of the ligands and modify the value of potential surface. I would suggest repeating the experiment with a naked nanostructure and less mobile ligands, proving then that facial recognition software is capable of identifying the right points. In conclusion, the idea is fascinating, but the results are not conclusive – distances of 2Å are substantial.

We thank all the Referees for their comments, criticism and suggestions, which have helped us to make the paper stronger and clearer to the readers. Below we repeat the comments by the referees (italics) and give our response below each comment.

Referee 1:

This paper describes the scanning tunnelling microscopy imaging (STM) of a silver cluster (Ag₃₇₄). To the best of my knowledge there is no report of such things to date. As such I think this paper should be published but in a specialised journal or a broad journal with reporting functions as Scientific Reports.

We agree with the Referee that our work is the first one where quite large, but atomically precise ligand-stabilized metal nanoclusters have been investigated with STM. However, we respectfully disagree on the novelty of approach and on potential impact of our work, as explained below.

The significance of this paper as such is minimal. The STM does not add anything to what X-ray crystallography has shown so far also on the same cluster. In fact it requires strong support from calculation.

Previous STM work on nanoparticles covered by ligands has been mostly on colloidal gold nanoparticles, with inhomogeneous shape and size distributions. The novel approach in our work is to take a previously well-defined system (Ag₃₇₄ stabilized by tert-butyl-benzene thiols) whose total atomic structure is available from single crystal X-ray diffraction data, investigate such samples under STM at temperatures down to liquid He temperature, combine the interpretation of the results to atomistic state-of-art DFT simulations on the topography images (which can be done with great accuracy since the atomistic structure is known), and finally use a pattern recognition algorithm to correlate the measured data to simulations. Our work shows conclusively that (a) it is possible to reach sub-molecular resolution of the topography in such systems, and (b) the structure can be correlated on the atomistic model by using pattern recognition algorithm. Theory and simulation is thus an integral part of this approach, and the significance of pattern recognition and “artificial intelligence” tools can only be predicted to grow in the future. The work thus demonstrates a new approach to the imaging problem of organic monolayers on curved nanoparticle surfaces; a problem that we think all Referees recognize as a very challenging one. As such it complements nicely some emerging non-direct experimental methods to study the structure of the ligand shell in such systems, like the most recent paper from Stellacci group, discussing the SANS method (Nature Comm 9 April 2018, citation added). We also think that the use of pattern recognition algorithms coupled with simulations to interpret experimental topography data of nanostructures is a truly new approach at this point and should prove useful to the community interested in accurate imaging (STM or AFM) of nanostructures, and our paper can be regarded as a pioneering work in this respect. As a supplementary material to this paper, we have in fact decided to release our house-made algorithm that performs the pattern recognition of STM images (or more generally, analysis of any topographical data with a set of well-defined extremal points). We think that this will significantly raise the impact of this work in the community interested in imaging (STM, AFM) of nanostructures.

The STM itself has been widely published on nanoparticles by the group of Stellacci. The authors do reference a controversy there but do not comment on it and neither add to it. The approach used is almost identical to the one described by such group in Ong et al ACS Nano (non cited), and the results achieved are similar to the ones described in the same paper and in Moglianetti et al. (not cited). Their minimal difference is that they achieved these results in liquid nitrogen and helium temperature, but low temperature results were described in Biscarini et al. (not cited).

We have cited a few previous papers in the introduction (and now added the three papers mentioned by the Referee) to illustrate the challenge of determining the structure of the organic monolayer on curved nanoparticle surfaces; a problem that we think all Referees recognize as a very challenging one. Our work is not aimed at resolving any existing controversies in the literature and not even contribute to existing debates, rather our work is a demonstration of another (novel) approach as we just explained above. Motivated by questions of Referee 2, we have also added (at the end of the paper) new discussion regarding the temperature-effect in imaging, based on our new DFT calculations on rotational barriers of t-butyl group in TBBT, and believe that this is also informative and a valuable addition in the paper.

Referee 2:

The present manuscript presents a study that combines experiment and atomic scale modeling to resolve the origin of features on a submolecular length scale in the experimental topography maps of Ag nanoparticles. The study is well motivated and addresses a topic that is of interest for a wide research community. The approach itself combines state-of-the-art experimental and computational methods in a novel fashion.

I think this work could be suitable for Nature Communications if the authors can comprehensively address the following issues:

We appreciate that the Referee sees the importance and potential impact of our work to a wider research community, and thank for his/her constructive comments.

I read the manuscript three times. Only upon the second reading did I see the main point. Hence I think the message should be sharpened still. For example it would help to avoid vague terms such as "facial recognition", which do not carry a particular meaning (at least not on first reading).

We appreciate this comment and have tried to make the main message of the work more clear in the revision, also defining better what we mean by "facial recognition".

The term "database" is used in multiple places. I assume this refers to the 1665 different orientations for which STM images for calculated? The authors also allude to a set of data for "flat" surfaces. It is unclear to me how these hang together.

We have now made the discussion clearer. We have additional experimental data on TBBT layers on flat Au(111) surfaces, as well as STM topography simulations by DFT methods on the appearance of one TBBT molecule on such surface. We agree that the use of "database" might have been confusing, and have now discarded that term.

"Pattern recognition" is a wide field in machine learning that has produced a wealth of sophisticated algorithms, many of which are readily available via well maintained libraries. By

comparison the algorithm employed here is quite simplistic. There is for example no statistical analysis of the reliability of the predictions. Moreover the minima in the generalized distance plotted in Figure 4D are not very pronounced. Hence, I am concerned how transferable this analysis is and how reliable the thus identified structures ought to be. What do I actually learn when looking at Figures 4I-L?

These are relevant comments and questions which prompted us to look for alternative codes/algorithms as reference to our house-made code for “pattern recognition”. In the time scale of this revision, we were able to compare our method to one publicly available software library, Open Source ComputerVision (OpenCV). We found that the tools offered by OpenCV for image recognition by identifying areas with uniform contrasts do not perform well for STM topography data analysis. Ideally, the most reliable experimental data with least effects from tip convolution can be expected from the central parts of the cluster where the contrasts are usually the smallest. We found that the OpenCV tools could not identify easily topographic features in that area and did not bring added value to our simple algorithm to identify a number of extrema (maxima and minima in the intensity). Positions with maximum intensities are the relevant ones since they directly point to the positions of the methyl groups in the ligands. We have done additional statistical analysis on the reliability of our method and discuss it in the revised text. In any case, we appreciate these comments very much since this comparison has given us additional confidence on our method, which is now also distributed as an open source algorithm as part of this paper in the Supplementary Information section.

Finally, while the measurements are carried out at very low temperatures, the molecules investigated here still feature rather soft rotational modes, which could in principle be populated at these temperatures. The "final" STM images are obtained by compounding multiple measurements over a longer period of time. How much molecular motion/vibration/rotation can I expect at these low temperatures over such long time scales and how would that affect the results? I would feel more confident if the authors could include a discussion of thermal effects and some estimates along these lines.

This is a very relevant comment and we have discussed the potential effects of the dynamics now in the revised text. Additional DFT calculations (Fig. S9) predict that during the sample preparation phase at room temperature, particularly rotational dynamics is very likely to happen, but at LHe temperature the whole system is essentially “frozen” during the time it takes to scan a typical STM image. At LN2 temperature, the dynamics still prevails, but only blurs out the intensity maxima (see new Figure 5 in the main text). It is indeed essential to go to much below room temperature to get well-resolved intensity maxima (corresponding to individual CH₃ groups in t-butyl).

Referee 3:

The manuscript entitled “Real-Space Imaging and “Facial Recognition” of a Ligand-Protected Ag₃₇₄ Nanocluster at Sub-Molecular Resolution” by Häkkinen et al. provides an interesting tool to characterise metal nanostructures. They combined STM and computer simulation to characterise the Ag₃₇₄ particle. Both results, experimental and computational, were implemented in a mathematical algorithm to identify the facets even though ligands surrounded the particle. Although the results for facial recognition are not fully conclusive, the idea is original, and I believe it can lead to fast advances in the field of nano-science.

We appreciate that the Referee finds our work original and having impact for a broad audience.

The information provided in the manuscript and supplementary material is not enough to support the claims of “facial recognition”. It can be, perhaps, that the simulation model is not accurate and therefore the points founded in the facial recognition step do not entirely agree with the experiment. The tip of the STM may also influence the arrangement of the ligands and modify the value of potential surface. I would suggest repeating the experiment with a naked nanostructure and less mobile ligands, proving then that facial recognition software is capable of identifying the right points. In conclusion, the idea is fascinating, but the results are not conclusive – distances of 2Å are substantial.

Here we refer to our response above to Referee 2. We have re-defined the “generalized distance parameter”, rating the quality of each of the 1665 theoretical topography images in comparison to the reference experimental data, in a way that should be now more transparent. The numbers for the “best” theoretical images mean (Figure 4), in practice, that the difference to the experimental data can be largely explained by a different rotational configuration of the methyl groups in the headgroups of TBBT ligands compared to the one found in the reference crystal structure. According to our DFT calculations, rotational dynamics takes place easily during the sample preparation phase around room temperature, but not at the imaging LHe temperature. Our calculations also predict that still at LN2 temperature it is possible to resolve the intensity maxima of individual CH₃ groups of t-butyl, which is actually supported by our experimental data. We have added discussion on this in the revised manuscript and have given new data in the Supplementary Information (figure S9) and in new Figure 5 in the main text. The additional suggestion by the Referee, to prepare a totally new “naked nanostructure” with less mobile ligands and to study it by STM, was not practical to realise in the time frame of this revision, since it would involve developing a new synthesis strategy and an extensive set of new low-T STM experiments.

REVIEWERS' COMMENTS:

Reviewer #2 (Remarks to the Author):

The authors have provided a rather complete reply to the questions raised by the referees. I appreciate that they included the source code for their algorithm although PDF is probably not the optimal format for distribution. Some sample input files would have been useful as well. Nonetheless I feel that the manuscript can be accepted now.